# Two-Dimensional Shear-Wave Elastography of the Thyroid in Clinically Healthy Dogs in Different Age Groups

**DOI:** 10.3390/ani14111528

**Published:** 2024-05-22

**Authors:** Denise Jaques Ramos, Tamiris Disselli, Diego Rodrigues Gomes, Luiz Paulo Nogueira Aires, Stéfany Tagliatela Tinto, Diana Villa Verde Salazar, Mariane Magno Ferreira Pereira, Brenda Santos Pompeu de Miranda, Ana Paula Luiz de Oliveira, Bruna Bressianini Lima, Ricardo Andres Ramirez Uscategui, Marcus Antônio Rossi Feliciano

**Affiliations:** 1Faculdade de Ciências Agrárias e Veterinárias, Universidade Estadual “Júlio de Mesquita Filho” (FCAV/Unesp), Jaboticabal 14884-900, São Paulo, Brazil; luiz.aires@unesp.br (L.P.N.A.); brenda.spme@gmail.com (B.S.P.d.M.); apl.oliveira@unesp.br (A.P.L.d.O.); bbressianinilima@gmail.com (B.B.L.); 2Faculdade de Zootecnia e Engenharia de Alimentos, Universidade de São Paulo (FZEA/USP), Pirassununga 13634-900, São Paulo, Brazil; tamiris.disselli@gmail.com (T.D.); rodrigues.gomes@usp.br (D.R.G.); stefanytinto@yahoo.com.br (S.T.T.); dsalazar.vet@gmail.com (D.V.V.S.); marianemagno@usp.br (M.M.F.P.); 3Faculdad de Medicina Veterinaria y Zootecnia, Universidad del Tolima, Ibagué 730006299, Tolima, Colombia; ramirezuscategui@hotmail.com

**Keywords:** dogs, diagnostic imaging, elastography, stiffness, thyroid, ultrasound

## Abstract

**Simple Summary:**

Shear-wave elastography is a technique based on ultrasound that assesses the stiffness of tissues and different organs in the body. The technique has recently been introduced to veterinary medicine, but not all organs have been studied in animals, and there is still limited knowledge about its applicability. The objective of this study was to evaluate the thyroid gland of clinically healthy dogs using 2D shear-wave elastography to establish qualitative and quantitative parameters of thyroid stiffness in dogs in different age groups. The healthy thyroid gland is characterized by a blue-color elastogram, indicative of low stiffness. There is no distinction between adult and elderly dogs, genders, or sizes.

**Abstract:**

The thyroid of dogs has not been extensively studied in 2D shear-wave elastography, making it challenging to apply this technique in the diagnosis of thyroid diseases in a non-invasive manner. The aim of this study is to evaluate the thyroid glands of healthy dogs using 2D shear-wave elastography in order to establish qualitative and quantitative parameters of tissue stiffness in dogs in different age groups. A total of 31 dogs of various breeds, sexes, and sizes were evaluated. Animals with clinical signs or ultrasound findings indicative of endocrine disease or thyroid lesions were excluded from the study. The shear-wave velocity data in meters per second (m/s) and color elastograms were evaluated and calculated using QelaXto™ 2D software. A healthy thyroid exhibits a blue-color elastogram, indicative of low stiffness. The reference range for the shear-wave velocity of thyroid tissue assessed by 2D shear-wave elastography can be between 1.6 and 2.0 m/s, with a variation of ±0.889 in adult and senior dogs.

## 1. Introduction

The thyroid gland is regarded as the most crucial of the endocrine glands in regulating metabolism, as it is responsible for the proper coordination of the body’s physiological processes through the secretion of hormones [1]. Injuries to the endocrine system and its components are characterized by functional disorders and pathological changes that manifest themselves in different systems of the animal [2]. These lesions may include degeneration, necrosis, vascular disorders, immune-mediated or infectious inflammation, atrophy, hyperplasia, and even neoplasia. These conditions result from or lead to instability in the function of the gland and the secretion of its hormones [3].

Clinical palpation can assist in the diagnosis of alterations associated with hyperplasia and neoplasms of the gland. However, it is not a reliable method for predicting tissue invasion and discrete thyroid alterations [4]. In order to confirm a diagnosis of thyroid alterations, it is necessary to measure the concentration of free tetraiodothyronine (free T4) and thyroid-stimulating hormone (TSH), perform imaging tests and collect material by biopsy or fine-needle aspiration biopsy (FNAB) [5,6,7].

A number of studies have demonstrated that elastography is a more effective technique than ultrasound in differentiating the characteristics of benign and malignant nodular lesions and presenting good diagnostic applicability [8,9]. Elastography has been investigated in a range of veterinary medical contexts. The technique has yielded promising results in the diagnosis of different alterations in internal organs, exhibiting high sensitivity and accuracy [10,11,12,13]. Despite the optimistic scenario, further studies are required to validate the reliability and relevance of shear-wave elastography of the thyroid as a diagnostic method for thyroid pathologies. Against this background, we discuss the potential use of shear-wave elastography as a complementary diagnostic technique for thyroid pathologies, due to its perceived advantages relative to other established methods.

Therefore, shear-wave elastography can be easily included in the protocol for cervical ultrasound examinations to elucidate different thyroid pathologies, provided that the technique is well consolidated in the evaluation of the thyroid. The aim of this study was to evaluate the thyroid of clinically healthy dogs of different sizes (large, medium, and small), sexes (females and males), and age groups (adults and seniors) using B-mode ultrasound and 2D shear-wave elastography (2D-SWE) in order to obtain qualitative and quantitative data on tissue morphology and stiffness. Furthermore, the objective is to establish a reference interval for the shear-wave velocity of thyroid tissues in adult and senior dogs.

## 2. Materials and Methods

### 2.1. Animals

The study was conducted using animals from kennels and their owners in the city of Jaboticabal, Brazil. A total of 31 canines were evaluated, comprising 11 males and 20 females of different breeds, with a weight range of 3 to 30 kg and varying sizes, including small, medium, and large dogs.

The animals were divided into two experimental age groups. The first group, designated as age group 1, consisted of adult dogs aged between 1 and 6 years old (n = 17). The second group, designated as age group 2, consisted of senior dogs aged seven years old or more (n = 14). The mean age and standard deviation of age groups 1 and 2 were 3.28 years (σ = 1.56) and 9 years (σ = 2.11), respectively. The groups were composed of individuals belonging to different breeds. The breeds represented in age group 1 were the Border Collie (2), French Bulldog (1), Pinscher (1), American Pit Bull (2), and mixed-breed dog (10). In age group 2, the breeds were as follows: the Dachshund (1), Golden Retriever (1), American Pit Bull (1), Poodle (3), and mixed-breed dog (8).

### 2.2. Clinical and Ultrasound Evaluation of Dogs

In order to ascertain the health of the selected dogs, the animals were previously submitted to anamnesis, general physical examination, thyroid palpation, and B-mode cervical (thyroid-focused) and abdominal ultrasound examinations. Animals exhibiting clinical signs or ultrasound findings indicative of endocrine disease or thyroid damage were excluded from the study. The following clinical signs were considered exclusion criteria: the presence of skin lesions accompanied by itching, areas of alopecia, polydipsia, polyphagia, sensitivity to cold, apathy, hyperactivity, and rapid weight loss or gain. In addition, the presence of the enlargement of organs or swelling of cutaneous/subcutaneous structures was also considered an exclusion factor. The B-mode ultrasound findings of the thyroid that were considered exclusion criteria included irregularity of the surface of the thyroid capsule, parenchyma with a coarse echotexture or mixed echogenicity, the presence of nodular formations regardless of their characteristics, and the presence of areas of fibrosis. Animals with ultrasonographic findings of endocrine alterations such as increased liver echogenicity, premucocele or biliary sludge in the gallbladder, decreased thyroid or adrenal echogenicity, increased thyroid or adrenal dimensions, thyroid nodules, and points of dystrophic calcification were excluded from the study. Dogs with evidence of inflammatory alterations, neoplasm, or increased dimensions of internal organs on B-mode ultrasound in the abdominal organs were excluded from the study. Age-related ultrasound alterations were not considered an exclusion criterion, for example, renal senility. Additionally, animals with agitated behavior that precluded the completion of imaging tests were excluded from the study.

All dogs with no alterations in their anamnesis, physical examination, or ultrasound examination and considered to be clinically healthy were included in the study. The ultrasound and elastography exams were conducted by the same examiner, and the images were evaluated by two imagers with a minimum of two years of experience.

For the ultrasound and elastography tests, the animals were subject to a comprehensive trichotomy of the ventral cervical region, extending from 2 to 3 cm below the angle of the mandible to the caudal cervical region (near the entrance to the thorax). Subsequently, the animals were positioned on a padded rail in dorsal decubitus, with their bodies parallel to the apparatus. The animals were physically restrained by their thoracic and pelvic limbs with the assistance of their owner and a trainee, without the need for sedatives and/or anesthesia. A water-based conductive gel, specifically designed for ultrasound examinations, was applied to the examination area in order to perform the techniques.

### 2.3. B-Mode Thyroid Ultrasonography

The B-mode ultrasound study of the thyroid was performed using the MyLab^TM^ X8 Platform equipment (Esaote, Firenze, Italy), with a 4–15 MHz multi-frequency and linear transducer. The settings for gain, frequency, focus, time gain compensation (TGC), and depth were adjusted according to the physical characteristics of each dog participating in the study. The lobes of the thyroid gland were identified in their typical location, situated between the sternocephalic and sternothyroid muscles, ventral to the trachea and caudal to the larynx. Both were evaluated in a longitudinal manner with regard to their dimensions (length and width) and echogenicity in relation to the adjacent muscles (hypoechoic, isoechoic, or hyperechoic) and surface regularity. The aforementioned characteristics were also assessed in the transverse section, except for the dimensions (Figure 1). Figure 2 presents the right thyroid lobe on B-mode ultrasound in a longitudinal section of a female dog.

### 2.4. Two-Dimensional Shear-Wave Elastography (2D-SWE) of the Thyroid

The 2D shear-wave elastography examination was performed subsequent to the B-mode ultrasound examination, utilizing the same device and transducer. The shear-wave velocity data in m/s and color elastogram were evaluated and calculated by analyzing the relative displacements of the tissue elements by an acoustic pressure pulse using the QelaXto^TM^ 2D software (Esaote, Italy).

Elastographic images were obtained in the longitudinal plane of each thyroid lobe, and a scale of 0.0 to 10.0 m/s was employed in all cases. Once the thyroid lobe was properly framed in the image, the QelaXto^TM^ 2D software quality map was initiated. The map indicated the most suitable locations for acquiring shear-wave velocity. These were colored green to indicate high quality (Figure 3), yellow to indicate medium quality, and orange to indicate low quality (Figure 4).

In all animals, shear-wave velocity measurements were only taken in high-quality regions, which were colored green. The color scale of the color elastogram ranged from blue (soft) to red (hard), passing through green and yellow (intermediate stiffness). To assess stiffness using shear-wave velocity, five samples of regions of interest (ROIs) were randomly selected, comprising different portions of the thyroid. These samples were selected using 0.2 cm diameter circles, as shown in Figure 5. The shear-wave velocity values of the analyzed portions were evaluated by the equipment’s software and expressed in m/s.

## 3. Results

The ultrasound and elastographic examinations of canines of different ages were conducted without any limitations and were easy to perform, with quick results regarding B-mode and elastographic characteristics. The data are presented as median ± interquartile range (IQR), and the significance value is defined as *p* < 0.005. The width, length, and depth values are presented in centimeters (cm), while the shear-wave velocity values are presented in m/s.

### 3.1. Adult and Senior Dogs

The width, length, and echogenicity of the thyroid gland on B-mode ultrasound did not demonstrate a significant difference between adult and senior dogs. The shear-wave velocity (SWV) values and depth between age groups demonstrated no significant difference in the elastographic evaluation. Despite this, all of the evaluated thyroids exhibited blue-color elastograms in the qualitative assessment, both in the group of adult dogs and the group of senior dogs. Table 1 presents the values for each age group for B-mode ultrasound and elastography characteristics.

### 3.2. Females and Males

The quantitative data obtained from B-mode ultrasound (width and length) and shear-wave elastography (shear-wave velocity and depth) revealed no significant differences between males and females of the species under investigation. This was evidenced by the p-value, which was greater than 0.005 in all aspects analyzed. Table 2 presents the values for each sex for the B-mode ultrasound and elastographic characteristics.

### 3.3. Small, Medium, and Large Dogs

To ascertain whether the size of the dogs influenced the quantitative B-mode ultrasound and elastographic characteristics, the animals were divided into three groups according to their size: large (L), medium (M), and small (P). The large size group was composed of dogs with an approximate body weight of between 25 and 30 kg. Dogs in the medium-sized group weighed between 15 and 24 kg, while those in the small-sized group weighed between 3 and 14 kg. A significant difference and positive correlation were found between size and width (*p* = 0.0031), length (*p* = 0.004), and depth (*p* = 0.0007) between the group of large-sized dogs when compared to the medium-sized and small dogs. The difference in shear-wave velocities between the thyroid lobes of dogs of varying sizes was not statistically significant. Table 3 presents the data for each size for the B-mode and elastographic ultrasound characteristics.

### 3.4. Statistical Analysis

The statistical analyses were carried out using R software version 3.3.0 (R Foundation for Statistical Computing, Vienna, Austria). Initially, the variables collected for the study were subjected to a series of statistical tests to ascertain their distribution and variance characteristics. These tests included the Shapiro–Wilk test for normality and Barlett’s test for homoscedasticity. The aforementioned tests demonstrated that the elastographic variables exhibited a non-parametric distribution. Subsequently, the repeated measurements or ROIs of the shear-wave velocity (SWV) and depth of each of the thyroid gland lobes assessed in all the animals were compared using the Friedman test. Statistical similarity was found between the SWV and depth of the right (*p* = 0.9767; *p* = 0.3154) and left (*p* = 0.7406; *p* = 0.599) ROIs. Consequently, the median SWV and depth of evaluation of the right and left portions were employed.

Subsequently, aspects like width, length, SWV, and depth of the right and left portions were compared using the Wilcoxon test for paired samples. The results revealed similarity between the portions for width (*p* = 0.1550), length (*p* = 0.8734), SWV (*p* = 0.2966), and depth (*p* = 0.5453), so the median of these variables was used. To accomplish this, the quantitative variables were compared among the age groups (adults and seniors), sexes (males and females), and sizes (small, medium, and large), and 95% median confidence intervals (95% CIs) were constructed as the normal ranges of the objective variables of the study when the results indicated similarity. For the statistical analysis of echogenicity, values were assigned to three different types of tone in comparison with the adjacent musculature. These values were as follows: hyperechogenic = 3, isoechogenic = 2, and hypoechogenic = 1.

The statistical analysis of the data focused on the age, sex, and size of the dogs. The objective was to correlate the variables studied with each other in order to identify potential correlations between them. The variables compared were age group, age, size, echogenicity, width, length, SWV, and depth.

The age group only showed a positive correlation for the age factor (r-Spearman = 0.868; *p* = 0). The size of the animals showed a positive correlation with the ultrasound aspects of width (r-Spearman = 0.746; *p* = 0), length (r-Spearman = 0.722; *p* = 0), and depth (r-Spearman = 0.574; *p* = 0.001). The B-mode ultrasound characteristics of the thyroid, including width and length, demonstrated a positive correlation with each other (r-Spearman = 0.695; *p* = 0) and also when correlated with the elastographic depth data (r-Spearman = 0.56; *p* = 0.001 for width and r-Spearman = 0.499; *p* = 0.004 for length). The elastographic data of depth and shear-wave velocity (SWV) did not demonstrate a correlation, as evidenced by the r-Spearman coefficient of −0.271 and the *p*-value of 0.014.

## 4. Discussion

This study was able to characterize the thyroid glands of clinically healthy adult and senior dogs using 2D shear-wave elastography, both in terms of qualitative data (color elastogram) and quantitative data (shear-wave velocity and depth). The findings of this study will contribute to the establishment of 2D shear-wave elastography as a promising diagnostic tool for diffuse and focal thyroid pathologies in the future.

One of the principal distinctions between this study and previous investigations is the comparison of the elastographic characteristics of the thyroid between adult and senior dogs. The hypothesis of this study was that animals over seven years of age would exhibit greater stiffness than younger ones. This could be explained by the longer period of exposure of the thyroid gland to immunological/inflammatory challenges and the appearance of fibrosis in many organs of senior animals. In another study using humans, the authors posited that the stiffness of a tissue in humans generally increases with age [14]. However, the comparison of shear-wave velocity between adult and senior dogs revealed no significant difference between the groups studied. Consequently, age is no longer a determining factor in thyroid stiffness, as the shear-wave velocity values in senior dogs were statistically similar to those of adult dogs (Table 1, where *p* = 0.0743). A similar conclusion was reported when evaluating thyroid tissue from 176 human subjects, aged between 21 and 91 years [14]. Moreover, the authors of the study report that there was no effect of sex/gender on stiffness.

Table 2 presents the SWV values between the sexes in the canine species under investigation, with a *p*-value of 0.8534. These findings corroborate those in humans, indicating that sex does not affect the elasticity of thyroid tissue [14]. It can be concluded that age is not a significant contributing factor to thyroid stiffness, which allows for the establishment of a single reference interval to define the value of the shear-wave velocity in 2D shear-wave elastography of adult and senior dogs. In addition, this study builds upon the previous work where only adult dogs aged between 1 and 5 years were studied [15].

With regard to the shear-wave velocity value found for adult dogs (1.63 ± 0.675 m/s), it is similar to the value established in a previous study, which established a value of 1.57 ± 0.1 m/s for the thyroid tissue of adult Beagle dogs in 2D shear-wave elastography [15]. Given the variation in the racial composition of the study population and the observed similarity between the SWV values and those previously reported in Beagles, it can be reasonably inferred that the values can be extrapolated to different breeds and sizes of dogs. This is further supported by the data presented in Table 3, which demonstrated that the shear-wave value (SWV) exhibited no significant difference between large, medium, and small dogs (*p* = 0.7222).

The characteristics of low stiffness, low wave velocity value, and a blue-colored elastogram of the healthy thyroid using 2D shear-wave elastography are probably due to the cellular constitution of its tissue. Healthy thyroid tissue, with normal levels of hormone concentration and secretion, is composed of follicular and medullary parafollicular cells of endodermal and neural crest origin, respectively. [3]. It can be reasonably assumed that the thyroid tissue of clinically healthy dogs exhibits the elastographic characteristics observed in this study, given that its cellular composition is unaltered and therefore does not affect its internal rigidity. This correlation suggests that a healthy thyroid is, in fact, a soft tissue.

When affected by malignant neoplasms or nodules, the thyroid tissue can exhibit increased stiffness and appear reddish to orange in elastography [16,17,18]. Indeed, the quantitative values of shear-wave elastography are higher in malignant nodules comprising numerous fibrous interstitial components and sand-like calcified bodies [16]. The present study, which demonstrates the normality of elastographic findings, may serve as a standard of normality for future work on detecting abnormalities in dogs.

The elastographic depth characteristic for acquiring shear-wave velocity demonstrated a notable disparity between large dogs and medium and small dogs (Table 3). The depth of the 2D shear-wave elastography range is a crucial factor to consider when discussing potential artifacts that could impact the quality of the wave signal or the technique due to the patient’s anatomy. A previous study found that 3.7% of elastographic artifacts that render images uninterpretable are attributable to the patient’s anatomy [19]. This can result in the technique producing some artifacts in giant-breed dogs, where it is assumed that the thyroid may be located at a depth greater than 1.905 cm. Consequently, the increased depth can result in a lack of elastographic signal, which is another elastographic artifact [19,20]. To ascertain the veracity of this hypothesis regarding the presence of this artifact in giant-breed dogs, it would be necessary to conduct studies evaluating the applicability and reproducibility of 2D shear-wave elastography in these animals.

In this study, no correlation was found between depth and shear-wave velocity values. This indicates that depth does not affect the values of shear-wave velocity. This finding is consistent with a study that compared two different elastography systems for assessing liver fibrosis in humans. The authors of the study reported that when assessed by 2D-SWE, shear-wave velocity was not influenced by depth in the areas of fibrosis assessed [21]. However, in the ARFI elastography used in the same study, SWV values varied according to depth in the same areas assessed previously [21].

The thyroid lobes of dogs of different sizes, ages, and sexes showed homogeneous, hyperechogenic, or isoechogenic parenchyma in relation to the adjacent musculature, in accordance with the literature [22,23,24,25,26]. With regard to the significant difference in width and length between the different sizes of dogs in this study, this is due to the influence of the size of the animal on the dimensions of the gland, as previously reported [23]. The consistency of the results obtained from the B-mode ultrasound of the thyroid gland in healthy dogs lends support to the assertion that the technique has well-established characteristics for assessing the thyroid.

During the 2D-SWE evaluations of the thyroid, this study identified certain limitations that may hinder the acquisition of images and values in certain dogs. These limitations included the technique’s inapplicability in agitated animals, those with wheezing, and those that do not allow proper positioning to access the ventral part of the neck. A total of six dogs were excluded from the study due to those characteristics, as they made it challenging to obtain a satisfactory quality map for measuring shear-wave velocity. Additionally, it was not possible to include a group of young dogs (aged between 6 and 11 months) due to the limited availability of animals in this age group. Furthermore, in some examinations, interference from compression by the arterial pulse of the common carotid artery was observed, as the proximity of this vessel to the thyroid sometimes made it difficult to acquire a high-quality elastographic map for measuring SWV. However, this issue was resolved by obtaining a new image until the quality map was colored green. Moreover, this new acquisition did not affect the examination time in the 31 dogs evaluated in the study.

## 5. Conclusions

Through this study of the thyroid gland of clinically healthy dogs, it can be inferred that 2D shear-wave elastography is a relatively easy-to-perform technique that offers rapid results regarding the stiffness of the gland in the species studied. Furthermore, a healthy thyroid gland exhibits a blue-color elastogram, indicating low stiffness, and there is no discernible distinction between adult and senior dogs, sexes, or sizes.

The shear-wave velocity shows no statistical difference between the aforementioned characteristics. This permits the creation and utilization of a uniform confidence interval for adult and senior canines, regardless of sex or size. Consequently, the reference interval for the shear-wave velocity of thyroid tissue assessed by two-dimensional shear-wave elastography can be established as between 1.6 and 2.0 m/s, with a variation of ±0.889 in adult and senior dogs.

## Figures and Tables

**Figure 1 animals-14-01528-f001:**
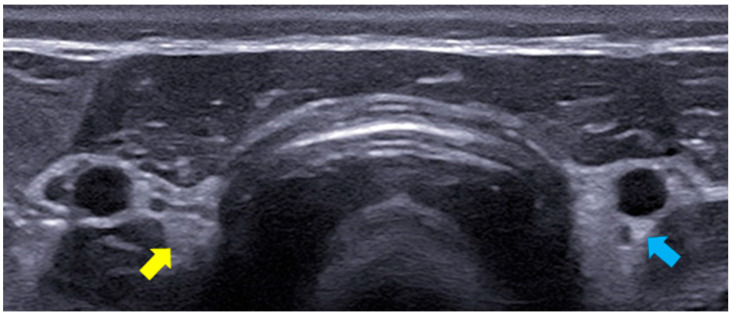
The B-mode ultrasound image of the right (yellow arrow) and left (blue arrow) thyroid lobes of a 9-year-old female dog.

**Figure 2 animals-14-01528-f002:**
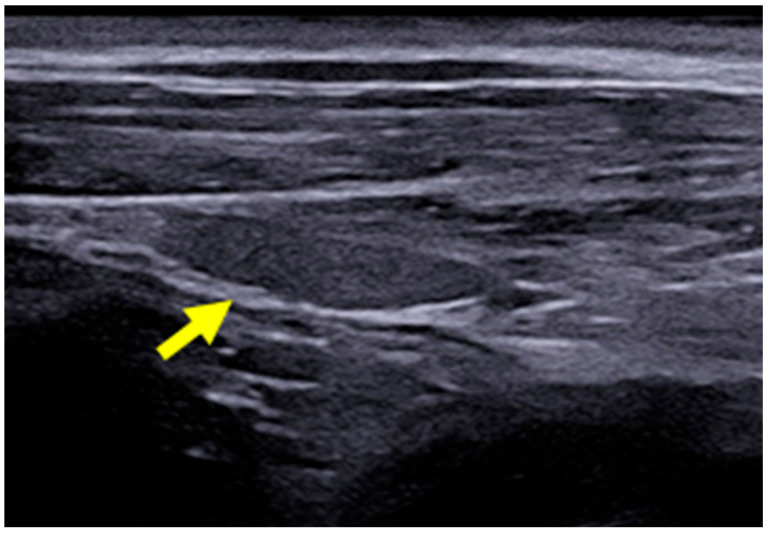
B-mode ultrasound images of the right thyroid lobe of a 3-year-old female dog (yellow arrow).

**Figure 3 animals-14-01528-f003:**
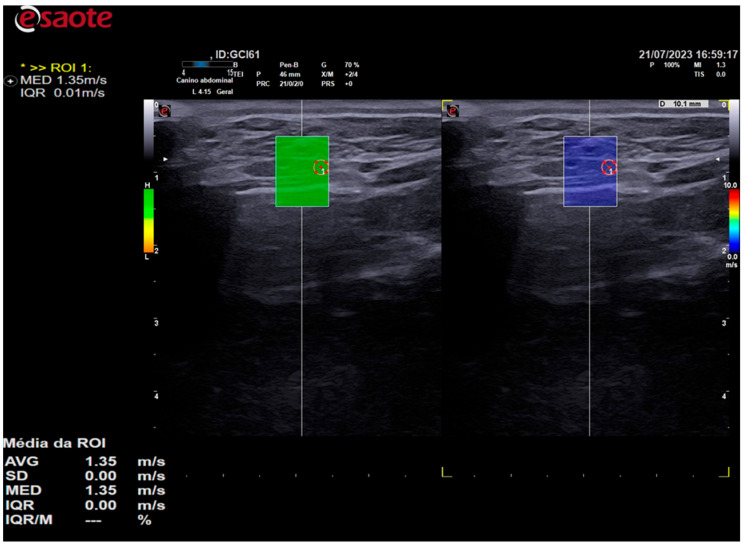
The image on the left is a 2D shear-wave elastography image of the left thyroid lobe of a 10-year-old dog. The elastographic image on the left shows a quality map from the QelaXto^TM^ 2D software, which is colored green. This indicates that the image is of high quality for measuring shear-wave velocity, which is a quantitative measurement. The right side of the elastographic image displays the colored elastogram (qualitative) from the QelaXto^TM^ 2D software in blue, indicating that the thyroid parenchyma had low stiffness (soft).

**Figure 4 animals-14-01528-f004:**
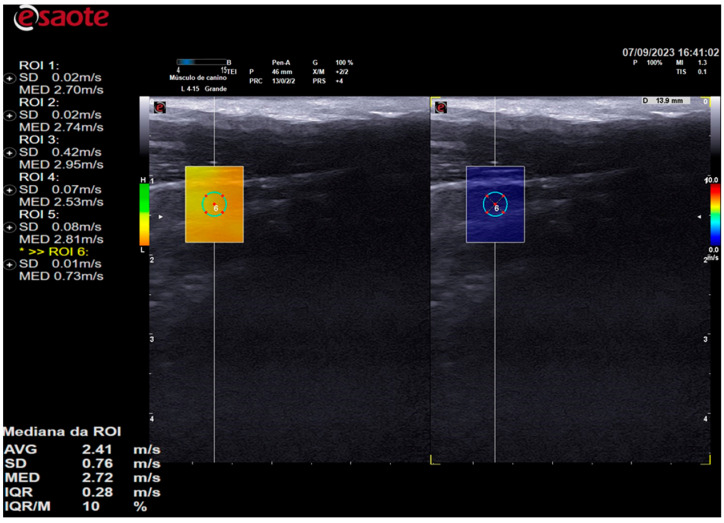
The image on the left is a two-dimensional shear-wave elastography image of the left thyroid lobe of a 3-year-old female dog. The elastographic image on the left shows the quality map from the QelaXto^TM^ 2D software, which is colored orange. This indicates that the quality is low for measuring shear-wave velocity (quantitative). The right side of the elastographic image displays a colored elastogram (qualitative) from the QelaXto^TM^ 2D software, colored in blue, indicating that the thyroid parenchyma exhibited low stiffness (soft).

**Figure 5 animals-14-01528-f005:**
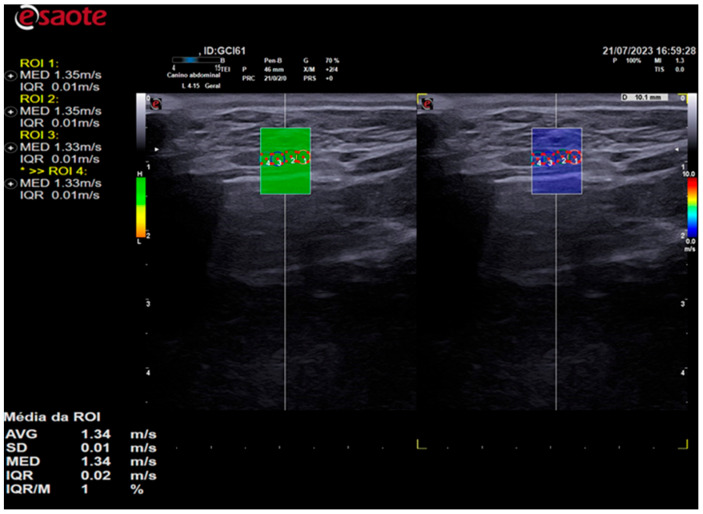
A 2D shear-wave elastography image of the right thyroid lobe of a 10-year-old dog is presented. The elastographic image on the left shows the quality map of the QelaXto^TM^ 2D software, which is colored in green and indicates high quality for measuring shear-wave velocity. Four ROIs were selected to measure shear-wave velocity, as indicated by the circles on the elastographic image. The right side of the elastographic image displays a color elastogram (qualitative) from the QelaXto^TM^ 2D software, colored in blue, indicating that the thyroid parenchyma had low stiffness (soft).

**Table 1 animals-14-01528-t001:** Comparison of width, length, echogenicity, shear-wave velocity (SWV), and depth between the age groups (AG) of adult (AG1) and senior (AG2) dogs.

Variable	AG	Median ± IQR	*p*-Value	95% CI
Width (cm)	AG1	0.335 ± 0.1675	0.6779	0.32
	AG2	0.345 ± 0.0988		0.39
Length (cm)	AG1	1.95 ± 0.993	0.6061	1.7
	AG2	1.922 ± 0.553		2.4
Echogenicity	AG1	3 ± 0	0.1333	-
	AG2	3 ± 2		-
SWV (m/s)	AG1	1.63 ± 0.675	0.0743	1.6
	AG2	1.877 ± 0.889		2.0
Depth (cm)	AG1	1.37 ± 0.547	0.2345	1.1
	AG2	1.1575 ± 0.6063		1.5

CI = confidence interval; IQR = interquartile range; SWV = shear-wave velocity; *p* < 0.005; cm = centimeters; m/s = meters per second.

**Table 2 animals-14-01528-t002:** Comparison of width, length, shear-wave velocity (SWV), and depth values between female and male dogs.

Variable	Gender	Median ± IQR	*p*-Value
Width (cm)	Female	0.3375 ± 0.16	0.6642
	Male	0.34 ± 0.105	
Length (cm)	Female	2.023 ± 0.905	0.1601
	Male	1.735 ± 0.475	
SWV (m/s)	Female	1.797 ± 0.798	0.8534
	Male	1.64 ± 0.415	
Depth (cm)	Female	1.3275 ± 0.75	0.7101
	Male	1.33 ± 0.475	

IQR = interquartile range; SWV = shear-wave velocity; *p* < 0.005; cm = centimeters; m/s = meters per second.

**Table 3 animals-14-01528-t003:** Comparison of width, length, shear-wave velocity (SWV), and depth values between large (L), medium (M), and small (P) dogs.

Variable	Size	Median ± IQR	*p*-Value	95% CI
Width (cm)	L	0.465 ± 0.1425 ^a^	0.0031 *	0.4235; 0.5645
	M	0.3375 ± 0.1175 ^b^		0.3202; 0.4198
	P	0.3075 ± 0.08 ^b^		0.2759; 0.3547
Length (cm)	L	2.858 ± 0.405 ^a^	0.004 *	2.282; 3,064
	M	1.99 ± 0.735 ^b^		1.845; 2.407
	P	1.72 ± 0.589 ^b^		1.522; 1.960
SWV (m/s)	L	1.63 ± 0.907 ^a^	0.7222	0.853; 2.517
	M	1.90 ± 0.971 ^a^		1.563; 2.739
	P	1.72 ± 0.678 ^a^		1.469; 2.400
Depth (cm)	L	1.905 ± 0.415 ^a^	0.0007 *	0.6888; 2.8572
	M	1.3425 ± 0.3513 ^b^		0.5794; 2.1126
	P	1.113 ± 0.482 ^b^		0.923; 2.135

CI = confidence interval; IQR = interquartile range; SWV = shear-wave velocity; *p* < 0.005; * significance value; cm = centimeters; m/s = meters per second; ^a^, ^b^ = indicate statistical difference by Tukey test (*p* < 0.05); equal letters do not differ significantly.

## Data Availability

The original contributions presented in the study are included in the article; further inquiries can be directed to the corresponding author(s).

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
