# Peer review of "Two-Dimensional Shear-Wave Elastography of the Thyroid in Clinically Healthy Dogs in Different Age Groups"

_animals, 2024, doi:10.3390/ani14111528_

Round 1

Reviewer 1 Report

Comments and Suggestions for Authors

Dear authors.

See comments end edits highlighted in the PDF attached. The Comments are always correlated to the highlighted text.

In general, the article is well written and easy to read and follow. My concerns is the lack of comparative groups. There were no differences in shear wave velocity in either of the animals as they were all healthy. How do you demonstrate the opposite without a group of animals with thyroid glands that show other features like increased velocity and hardening. I believe the study would need that comparative group to truly add value.

The ultranosography of the thyroid gland part including text and images should be reduced or partly removed as in not novel. I understand you used it to select you patients and to demostrate no illness but other than that add little value to the manuscript.

All the best

Author Response

May 3rd, 2024.
Dear Editors,
Re: animals-2986472
Thank you for your help to improve our paper and overall favorable assessment of the present submission. As recommended, this paper has been reviewed and subjected to a round of linguistic revisions. We have also revised our original manuscript taking into consideration all the reviewers’ comments and suggestions and as detailed below.
Referee(s)' Comments to Author: General comments “My concerns is the lack of comparative groups. There were no differences in shear wave velocity in either of the animals as they were all healthy. How do you demonstrate the opposite without a group of animals with thyroid glands that show other features like increased velocity and hardening. I believe the study would need that comparative group to truly add value.”
- The study did not include a group of animals with thyroid abnormalities because the objective was to ascertain the thyroid's behavior in clinically healthy animals, thus paving the way for future studies to be conducted on sick animals. “The ultranosography of the thyroid gland part including text and images should be reduced or partly removed as in not novel. I understand you used it to select you patients and to demonstrate no illness but other than that add little value to the manuscript”
- The part about ultrasonography of the thyroid gland was reduced as asked. “The writing as well as the English need moderate editing. Expressions such as, in general, should be avoided.”
- The english was revised and edited, and the corrections were made to follow the right grammatical. “I encourage the authors to clarify in future studies the role of elastography in the hypothyroidism and in euthyroid sick syndrome diagnosis.”
- This suggestion will be appreciated and considered for our research group in a future study. Figure 1 and Figure 2B
- The figure 1 was edited and it is confirmed that the arrows point to a hyperechoic, diminutive area (thyroid). Figure 2B was removed due to its poor quality. Figure 6
- The figure 6 was removed because its content is not novel and does not contribute to the manuscript's overall content. Abstract “Starting sentence with number”
- This issue is rectified.
“Exact sentence repeated above in summary”
- The sentence was removed of the summary and now it’s just in the abstract. “You should start with the background (Place the question addressed in a broad context and highlight the purpose of the study). You should consider rewrite the whole abstract”.
- The abstract was rewritten as asked. Introduction Line 48-49: “Don’t understand what do you mean here. Needs to know if there was supplementation given?”
- The mention to “thyroid hormone supplementation” was removed due to this misunderstanding. Line 50-59: “This paragraph needs to be better explained”
- The paragraph was reformulated as asked. Materials and Methods ‘Ethical statement’ place
- The “Ethical statement” was replaced as asked in the journal requirements Materials and Methods - Animals Line 72: “Wrong punctuation and commas make it difficult to read”
- The english was revised and the corrections were made to follow the right grammatical. Line 81: “What is SRD?”
- The “SRD” abbreviation was changed by “mixed breed dog” term, the authors noted that the SRD was not translated properly. Materials and Methods – Clinical and ultrasound evaluation of dogs Line 91-93: “Not sure what you mean. Enlargement of organs or swelling of cutaneous or subcutaneous structures?”
- The “volume increases in any region of the body” referred to both. The text was edited and we included the right terms to avoid misunderstanding. Line 97-100: “You should consider include in the list specific findings related with endocrine disease (especially hypothyroidism) e.g. aumented ecogenicity of the liver or patchy pattern, premucocele, distrophic calcifications.”
- The specific findings related with endocrine disease was included to the paragraph as asked. Line 106-107: “What is an alteration in a medical history? Needs to rephrase”
- The phrase was edited and the “medical history” was excluded. In the place we write “anamnesis”. Materials and Methods – B-mode thyroid ultrasonography Line 122: “Needs to expand always the first time is mentioned”, about TCG
- The “time gain compensation (TGC)” was add to the text to explain the abbreviation. Materials and Methods – 2D Shear Wave Elastography (2D-SWE) of the thyroid Line 142: “Is that the right spelling?”, about ‘slastogram’
- The typo of the word “elastogram” has been corrected. Materials and Methods – Statistical analysis “This part should be at the end of results?”
- We have placed “Statistical analysis” subsection to the end of results as asked. Results – Adult and senior dogs Line 193-194: “The way they are classified does not match. What is GE? And the mention of AG, AG1 and AG2 where is that in the table?”
- The “GE” abbreviation was changed by “AG”, the authors noted that the GE was not write properly. The AG, AG1 and AG2 was added to the table as well. Discussion Line 268-270: “The actual reason is that because of that there are fibrotic, post inflammatory changes?”
- Yes, and the text was edited to include this throught as asked.
Once again, I would like to thank you for the constructive criticism and useful suggestions. We trust that our revised version of the manuscript is now suitable for publication in the Animals. I have attached our re-editions to the text and figures legends and rewritten the topics of the manuscript. We also compromise in case of positive acceptance of the manuscript to send the same to a professional translator, with experience in veterinary manuscripts, specifically with our works, for a grammatical correction and revision of the text (work of flow and syntax) for later publication. Please do not hesitate to contact me directly if you require any further clarification.
Sincerely,
Marcus Antonio R. Feliciano

Reviewer 2 Report

Comments and Suggestions for Authors

   This study presents the basis for the use of shear wave elastography in dogs of different ages. The results in this new way are novel due to the variability of ages and breeds presented.

   The data have been collected meticulously and presented in an orderly and structured manner.

  The writing as well as the English need moderate editing. Expressions such as, in general, should be avoided.

   More bibliographical references should be included, Animlas recommends no less than 30, therefore, more references to shear wave elastography in human beings should be included.

I encourage the authors to clarify in future studies the role of elastography in the hypothyroidism and in  euthyroid sick syndrome diagnosis.

Abstract: You should start with the background (Place the question addressed in a broad context and highlight the purpose of the study). You should consider rewrite the whole abstract.

 99 “Dogs that showed inflammatory, neoplastic or increased dimensions on B-mode ultrasound in abdominal organs were not included in the study.”

You should consider include in the list specific findings related with endocrine disease (especially hypothyroidism) e.g. aumented ecogenicity of the liver or patchy pattern, premucocele, distrophic calcifications.

 Discussion

274 The results of this  study help to consolidate 2D shear wave elastography as a potential diagnostic method for diffuse and focal thyroid pathologies in the future.

Include references from human beings literature

313 When affected by neoplasms or malignant nodules, the thyroid can suffer increased  tissue stiffness, with a reddish to orange elastogram on elastography.

Reference needed.

336 However, in the ARFI elastography used in the same study,

ARFI (Acoustic Radiation Force Impulse)

References.

You should include this reference dealing with thyroid ultrasound in dogs.

Thyroid sonography as an effective tool to discriminate between euthyroid sick and hypothyroid dogs

Comments on the Quality of English Language

The writing as well as the English need moderate editing. Expressions such as, in general, should be avoided.

The constant use of "that" should be changed by passive sentences.

Author Response

(The authors gave the same response as above.)

Round 2

Reviewer 1 Report

Comments and Suggestions for Authors

Thank you for the corrections and the edits.

I think it looks and reads better. 

The tables (all of them) need to add units either in the top or the captions.

Otherwise if all corrected and grammar reviewed I am happy to proceed with the acceptance of the manuscript